# Disseminated Intravascular Coagulopathy Is Associated with the Outcome of Persistent Inflammation, Immunosuppression and Catabolism Syndrome

**DOI:** 10.3390/jcm9082662

**Published:** 2020-08-17

**Authors:** Kensuke Nakamura, Kentaro Ogura, Hidehiko Nakano, Hiromu Naraba, Yuji Takahashi, Tomohiro Sonoo, Hideki Hashimoto, Tadahiro Goto

**Affiliations:** 1Department of Emergency and Critical Care Medicine, Hitachi General Hospital, 2-1-1, Jonan-cho, Hitachi, Ibaraki 317-0077, Japan; be.rann1988jp@gmail.com (H.N.); nrbhrm@gmail.com (H.N.); yuji.mail@icloud.com (Y.T.); sonopy77@gmail.com (T.S.); hidehashimoto-tky@umin.ac.jp (H.H.); 2Faculty of Medicine, The University of Tokyo, 7-3-1 Hongo, Bunkyo, Tokyo 113-0033, Japan; mathbass.ken531@gmail.com; 3TXP Medical Co. Ltd. 3-13 Nihonbashiyokoyamacho, Chuo-ku, Tokyo 103-0003, Japan; tadahirogoto@gmail.com; 4Department of Clinical Epidemiology and Health Economics, School of Public Health, The University of Tokyo, Tokyo 113-0033, Japan

**Keywords:** coagulopathy, CRP, DIC, PICS, PIICS, sepsis

## Abstract

Persistent inflammation, immunosuppression and catabolism syndrome (PIICS) often occur after critical care. Disseminated intravascular coagulation (DIC) is expected to be associated independently with PIICS development. We retrospectively analyzed 5397 patients admitted to the Hitachi General Hospital emergency and critical care center during four years. We classified PIICS as C-reactive protein > 3.0 mg/dL or albumin < 3.0 g/dL or lymphocyte count < 800/μL on day 14. Prolonged hospital stay (>14 days) without PIICS and early recovery (discharged alive within 14 days) were assigned as non-PIICS. Early death (death within 14 days) was identified. We analyzed the association between the International Society on Thrombosis and Haemostasis overt DIC and PIICS outcomes. Results revealed 488 PIICS, 416 early death and 4493 non-PIICS cases. Analyses showed DIC as associated significantly with mortality, the Barthel index at discharge and PIICS development. Multivariate regression analysis and a generalized structural equation model identified DIC on admission as an independent risk factor for PIICS in surviving patients.

## 1. Introduction

Persistent inflammation, immunosuppression and catabolism syndrome (PIICS) is prolonged inflammation associated with immunodeficiency and catabolism, often affecting patients after critical care [1,2]. Interaction between myeloid-derived suppressor cells and inflammatory cytokines, such as interleukin (IL)-6 and IL-8 with elevated C-reactive protein (CRP), are regarded as key elements for PIICS [3]. When broadly dividing the clinical courses of intensive care, three courses of critical conditions and treatments become apparent over the long term: early death, early recovery and chronic critical illness (CCI) [4,5]. Actually, PIICS is regarded as one component of CCI. Once PIICS develops, other pathogens can easily infect the patient, activities of daily living (ADL) is worsen and their long-term mortality is eventually increased [1,2].

The definition of PIICS was originally proposed as prolonged hospitalization > 14 days, CRP > 0.15 mg/dL, total lymphocyte count < 800/mm^3^, weight loss of > 10% or body mass index < 18 during hospitalization, Creatinine height index < 80%, albumin < 3.0 gm/dL, pre-albumin < 10 mg/dL and retinol binding protein < 10 μg/dL^1^ (Table 1). The original reference materials do not describe whether these conditions needed to be filled in as “and” or “or”. However, prolonged stay and CRP are expected to be most important for PIICS leading to persistent inflammation [1,2]; other factors can be “or” conditions. Moreover, this CRP value was apparently too low. An earlier report described day 14 CRP as an equally suitable criterion to 3.0 mg/dL for PIICS [6]. With such clinical criteria to detect PIICS development, the patients’ immunosuppression can be evaluated before actual infection occurs. The catabolism can be evaluated for eventual physical function and muscle volume.

Sepsis is a main condition causing PIICS [2]. However, other risk factors for PIICS have not been clarified in the literature. Herein, disseminated intravascular coagulation (DIC) shows a syndrome with continuous coagulopathy in systemic blood vessels, presenting with multiple micro-thrombosis and bleeding [7]. Extensive cross talk between coagulation and inflammation, by which activation of one system amplifies activation of the other, can induce tissue damage and organ failure [8]. Therefore, coagulopathy and DIC can affect PIICS development via persistent inflammation [9]. Unfortunately, no report of the relevant literature describes a clinical study analyzing correlation between coagulopathy and PIICS development.

We hypothesize that DIC is an independent risk factor for PIICS development. To assess that hypothesis, inpatient data of our emergency and critical care center were extracted and analyzed retrospectively. Along with the diagnosis of DIC using International Society on Thrombosis and Haemostasis ISTH criteria, PIICS is classified according to its day 14 criterion. Its correlation to coagulopathy were analyzed.

## 2. Materials and Methods

### 2.1. Dataset and Cohort

Data of patients admitted to Hitachi General Hospital emergency and the critical care center ICU or emergency ward from April 2016 through May 2019 were analyzed for this study. This retrospective study was approved by our hospital ethics board (2017–2019). Coagulation analyses of patients’ plasma were performed on admission with opt-out patient agreements.

### 2.2. Classification of PIICS, Non-PIICS, Early Recovery and Early Death

For the classification of PIICS, we first extracted “early death” patients, who died within 14 days and “early recovery” patients, who were discharged alive within 14 days. The other population included patients for whom the hospital stay was >14 days.

Subsequently, PIICS patients were identified as CRP > 3.0 mg/dL or albumin < 3.0 g/dL or lymphocyte counts of <800/μL on day 14. CRP, albumin, and lymphocyte counts were extracted with the nearest measurements to day 14 within days 11–17. When the patients had no value for each factor, it was applied as negative in an or-conditional expression. “PIICS” was inferred if one or more of three components were matched. Patients were classified as “prolonged hospital stay without PIICS” if all three components were negative.

Finally, early recovery and prolonged hospital stay without PIICS were categorized as “non-PIICS”.

### 2.3. Other Outcome Evaluation

As patient characteristics, the following were extracted and evaluated: age, sex, DIC, sequential organ failure assessment SOFA score and the acute physiology and chronic health evaluation (APACHE II) score on admission. Coagulation abnormalities were scored using ISTH overt DIC criteria and the Japanese diagnostic criteria of acute phase DIC (JAAM-DIC) (Appendix A) [10]. In ISTH overt DIC criteria, the fibrin/fibrinogen degradation products (FDP) were used for fibrin-related markers: no-increase, moderate increase, and strong increase were defined, respectively for FDP <10 μg/mL, 10–25 μg/mL, and ≥25 μg/mL [10]. As adjunctive therapies, mechanical ventilation, renal replacement therapy and extracorporeal membrane oxygenation ECMO use were evaluated. When the existence of sepsis was found from the patient files, the case was counted and recorded. We also counted post-surgery cases. As outcomes, along with the described outcomes of PIICS, non-PIICS and early death, we evaluated in-hospital mortality, length of hospital stay, Barthel Index at discharge (evaluated by nurses), CRP, albumin and lymphocyte count on day 14. As coagulation-related markers, platelet, FDP, prothrombin time (PT), activated partial thromboplastin time (APTT), fibrinogen and antithrombin activity on admission were evaluated. Hemoglobin Hb, creatinine and HbA1c concentrations on admission were also evaluated.

### 2.4. Statistical Analysis

Continuous variables are expressed as the mean ± standard deviation and are compared using Student’s *t*-test or one-way analysis of variance when the null hypothesis is not rejected by the Shapiro–Wilk test. Continuous variables are expressed as the median (interquartile range) and are compared using Mann–Whitney U testing or Kruskal–Wallis tests when the null hypothesis is rejected by the Shapiro–Wilk test. Tukey–Kramer or Steel–Dwass tests were applied to examine differences of pairs in the multigroup. For categorical variables, the proportions of patients in the respective categories were calculated. Then the groups were compared using chi-squared tests or Bonferroni tests of multigroups. Nominal logistic regression analysis was applied to elucidate PIICS-in-PIICS and non-PIICS patients, excluding early death. Univariate logistic regression analysis was conducted first in each factor. Then, multivariate logistic regression analysis was performed with the factors. To check multicollinearity between independent variables, the variance inflation factor was calculated before performing multivariate logistic regression. Multicollinearity was regarded as present when the variance inflation factor was greater than 10.

To elucidate the relative, complex association between factors and PIICS, we used a generalized structural equation model (GSEM) [11,12]. The model structure was developed based on prior knowledge [9,13,14,15]. Although numerous correlated variables were included, we assumed that they reflect only a few pathophysiological processes. To restrict the number of variables, we used the following variables as surrogates of the pathophysiological processes: hemoglobin, serum creatinine (renal dysfunction), APACHE II score (patient severity), serum albumin (nutritional status), C-reactive protein (inflammation) and the ISTH overt DIC score (coagulopathy) on the first day of admission. Although these pathophysiological processes may partially explain the underlying associations in PIICS, the PIICS development could not be explained solely based on the initial conditions. Therefore, some “other” pathway to the development of PIICS is expected to exist. Consequently, we used the “other” factor as a latent variable related to PIICS. The path coefficient values correspond to the standardized solution of the model. As such, they allow for direct comparison among the strengths of paths in the model.

GSEM analysis was done with R (R 3.6.1). The other statistical analyses were conducted using JMPpro14 (SAS Institute, Inc.). Results for which *p* was less than 0.05 were inferred as significantly different *.

## 3. Results

The study outline for patient extraction flow is depicted in Figure 1. A total of 5397 patients were admitted to our emergency and critical care center icu/emergency ward during this period. Of those patients, 416 died within 14 days (early death); 4248 were discharged alive (early recovery). Of the remaining 733 patients, for whom the hospital stay was > 14 days, 488 PIICS were identified as having CRP > 3.0 mg/dL or albumin < 3.0 g/dL or lymphocyte count < 800/μL on day 14. The other 245 patients were designated as having prolonged hospital stay without PIICS. Among the 733 patients who remained hospitalized longer than 14 days, 618 were found for whom day 14 CRP was obtained and distributed as mean 4.6 ± 5.5, median 2.75 (0.9, 6.5).

Data of 5397 patients admitted to the ICU/emergency ward were analyzed. Of these patients, 416 patients died within 14 days (early death); 4248 patients were discharged alive (early recovery). Of the remaining 733 patients, for whom the hospital stay was > 14 days, 488 PIICS were identified as CRP > 3.0 mg/dL or albumin < 3.0 g/dL or lymphocyte counts < 800/μL on day 14; the other 245 patients were assigned as having Prolonged hospital stay without PIICS.

CRP; C-reactive protein, PIICS; persistent inflammation, immunosuppression and catabolism syndrome.

Table 2 presents basic characteristics found with and without ISTH overt DIC on admission. Patients with DIC were of higher age, higher severity and ICU admission rate and more frequent mechanical ventilation, renal replacement therapy and ECMO use (*p* < 0.0001, respectively). Patients with sepsis were more numerous in the ISTH overt DIC group (*p* < 0.0001). Initial CRP and albumin on admission were worse in patients with DIC than in non-DIC patients (*p* < 0.0001). Table 3 presents those outcomes. In-hospital mortality was significantly higher in ISTH overt DIC patients than in non-ISTH overt DIC patients (25.2% vs. 6.8%; *p* < 0.0001). The length of hospital stay was longer in the ISTH overt DIC group; the Barthel index at discharge (in patients discharged alive) was also significantly lower in the ISTH overt DIC group (*p* < 0.0001, respectively). Moreover, for ISTH overt DIC vs. non-ISTH overt DIC, median CRP on day 14 was higher (3.6 vs. 2.1 mg/dL, respectively; *p* < 0.0001); mean albumin on day 14 was lower (2.3 vs. 2.5 g/dL, respectively; *p* = 0.0015). In particular, PIICS and early death outcomes were remarkably greater in the ISTH overt DIC group; early recovery occurred more often in the non-ISTH overt DIC group (*p* < 0.0001).

Then, correlation between laboratory findings related to coagulopathy and PIICS outcomes were analyzed (Table 4). Significant difference was found between the groups in platelet, FDP, PT, APTT, fibrinogen and antithrombin activity on admission. It is noteworthy that both the ISTH-overt DIC and JAAM-DIC rate were higher and that the antithrombin activity was lower in the PIICS and early death groups than in non-PIICS, which suggests that aggressive coagulopathy is associated not only with early death, but also with PIICS as the outcomes.

To investigate baseline risk factors for PIICS, we applied univariate and multivariate regression analysis for PIICS for patients in whom early death was excluded (Table 5). Univariate regression analysis results indicate significant correlation to PIICS development by all of the following: age, male sex, SOFA and APACHEII score, sepsis, initial CRP, albumin, lymphocyte count, Hb and creatinine on admission. Furthermore, DIC and antithrombin activity were indicated as significant factors for PIICS. We confirmed important factors without multicollinearity. Subsequently, we analyzed them using multivariate regression analysis as shown in Table 5. Finally, age, male, APACHE II, sepsis, albumin, lymphocyte counts, Hb, creatinine, HbA1c and ISTH overt DIC on admission were regarded as independent risk factors for PIICS. The same regression analysis, but with application limited to sepsis patients, produced the results presented in Appendix A. Almost identical results were obtained from univariate regression analysis. However, only age, male, APACHE II score, albumin and Hb were confirmed as independent factors for PIICS.

Regarding the GSEM, our model shows good fit to explain the association of interest: goodness-of-fit, 0.98; adjusted goodness-of-fit, 0.92; root mean square error of approximation index, 0.08 (95% CI 0.07–0.08); standardized root mean residual, 0.045. The association and corresponding standardized coefficients are shown in Figure 2 and in Appendix A. This analysis revealed inflammation and nutritional status and inflammation and coagulopathy as mutually correlated to a great degree under characteristics including severity and infection. The degrees of CRP (inflammation), ISTH overt DIC score (coagulopathy) and nutrition status (albumin) on the first day were found to be important factors for PIICS development (standardized coefficient, 0.17, *p* < 0.001 and 0.09, *p* < 0.001, respectively). However, other factors, perhaps via overall severity, have the most significant effect on PIICS development (standardized coefficient, 0.75; *p* < 0.001).

A generalized structural equation model is described for the complex association between factors and PIICS. Inflammation and coagulopathy on the first day were found to be important factors for PIICS development. However, another factor, perhaps via overall severity, was found to have the greatest effect on PIICS development. Solid lines represent a significant relation with *p* < 0.05. Dotted lines represent lack of significant relation.

PIICS; persistent inflammation, immunosuppression and catabolism syndrome.

## 4. Discussion

This study found DIC to be associated not only with early death, but also with PIICS development in surviving patients. Furthermore, DIC affects the hospital stay and ADL at discharge. Age, male, sepsis, albumin, lymphocyte count, Hb, creatinine, HbA1c and ISTH overt DIC on admission were identified as risk factors that were independent of disease severity for PIICS in surviving patients.

This report is the first of a study clarifying correlation between coagulopathy and PIICS based on large clinical data. Our results suggest that patients with DIC who are able to survive would die more frequently and that they would develop PIICS with greater likelihood. It has already been reported that DIC affected mortality and hospital stay length, possibly because of crosstalk with inflammation and consequent organ dysfunction [16,17]. Anti-coagulant therapy is used occasionally in clinical practice with the expectation of breaking this negative spiral [16,18]. Because PIICS and deteriorated ADL are apparent in patients with DIC, we propose that PIICS development be analyzed in surviving patients along with mortality as outcomes of intervention for DIC. We often experience another infection event as a second hit and observe ADL decline via catabolism after critical care. Detecting PIICS is important for formulating a broad definition of post-intensive care syndrome and for intervention as early as possible to prevent and mitigate PIICS [19]. Intervention against coagulopathy is anticipated as one approach to improve PIICS [9]. Further investigations to examine interventions are expected to be necessary for future study.

Coagulopathy is regarded as playing an important role in PIICS. Although the activation of coagulation is protective to counteract pathogens through fibrin deposition [20], excessive coagulopathy can be expected to accelerate inflammation [8] and to worsen microcirculation, possibly leading to multiple organ failure [7]. Reportedly, extreme coagulopathy has been observed in the blood and organs in PIICS model mice [21]. Based on results of the present study, decreased antithrombin activity was identified as a risk factor for PIICS. Because antithrombin dysfunction increases fibrin formation and insufficient fibrinolysis, which engenders microvascular thrombus, decreased antithrombin activity in DIC is reportedly correlated with poor prognosis [22]. The possibility exists that interventions such as antithrombin supplement and anti-coagulant therapy, which recover antithrombin function, are effective for PIICS prevention.

In our generalized structural equation model, initial nutrition status, inflammation and coagulopathy affected PIICS development. It was reasonable because PIICS was fundamentally the inflammation and catabolism syndrome. These factors were mutually correlated through crosstalk between inflammation and coagulopathy. They were also affected strongly by the overall disease severity and infection condition. Sepsis has been reported as one of the strongest PIICS development conditions [1,2]. Furthermore, as our multivariable regression analysis results have suggested, the background of age, male sex and diabetes and complication of renal dysfunction and anemia were associated with PIICS development. Comorbidities and higher age have been regarded as risk factors for PIICS [13]. In particular, kidney injury [3] and diabetes [23] were discussed as directly involved with PIICS development. Moreover, a recent study showed that anemia had been correlated with persistent inflammation in critically ill septic patients [24]. These comorbidities can be regarded as independent risk factors for PIICS.

Nevertheless, the strongest factor exacerbating PIICS development was neither comorbidity nor sepsis, but disease severity. The APACHE II score was the strongest independent risk factor from initial CRP and albumin: it was sufficient to affect PIICS criteria. Coagulopathy is another aspect of severity. It is interesting that the greater severity engenders more PIICS, independent of the background and initial laboratory findings.

This study included some limitations. First, this study was conducted as a single-center retrospective study. Regional and hospital characteristics may influence study results. Actually, many elderly patients in Japan are admitted to emergency and critical care centers. These study patients therefore included many elderly people. Second, this study included all patients who were admitted to our emergency and critical care center. Among them were patients with various diseases such as sepsis and post-surgery complications. To elucidate a disease-specific PIICS relation, the same analyses should be conducted with disease-limited patients in future studies. Third, some unmeasured or unknown confounding factors are expected to affect the multiple regression analysis. Actually, PIICS development cannot be explained solely by initial conditions and laboratory findings. Similarly, GSEM may have other confounding factors. Fourth, treatments after admission were not considered for this study. They may include some anti-coagulant therapy for DIC. Fourth, the CRP transition of this study includes a population for which treatment failure, including inadequate source control or other disease or infection, occurred during the clinical course. Therefore, PIICS and CCI are compound conditions including these treatment failures and second hits. Finally, CRP, albumin and lymphocyte counts on day 14 were allowed within days 11–17, which included deficits. Laboratory data should be collected accordingly on day 14 without deficits for future prospective studies.

## 5. Conclusions

Results indicate DIC as an independent risk factor not only for mortality, but also for PIICS development in surviving patients.

## Figures and Tables

**Figure 1 jcm-09-02662-f001:**
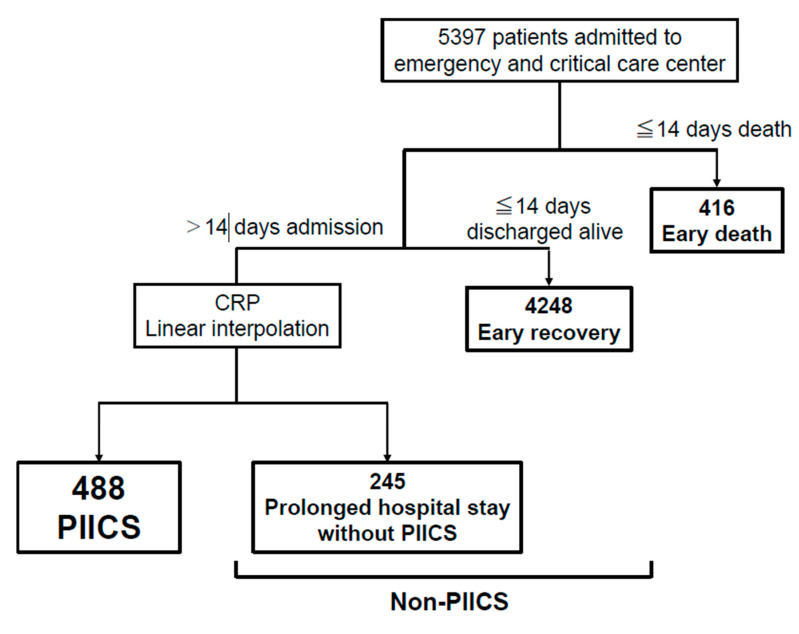
Study outline.

**Figure 2 jcm-09-02662-f002:**
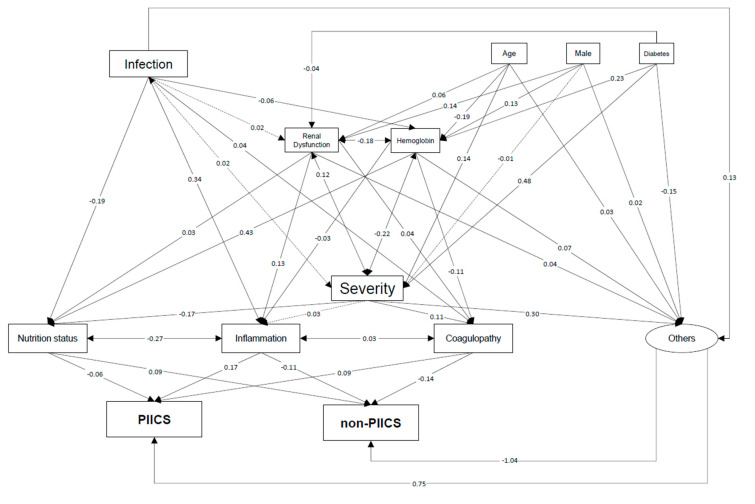
Generalized structural equation model for PIICS.

**Table 1 jcm-09-02662-t001:** Clinical definition of Persistent inflammation, immunosuppression and catabolism syndrome (PIICS).

Definition of PIICS	
Persistent	Prolonged hospitalization > 14 days
Inflammation	C-reactive protein > 3.0 mg/dL
Immunosuppression	Total lymphocyte count < 800/mm^3^
Catabolism	Weight loss of >10% or body mass index < 18 during hospitalization
	Creatinine height index < 80%
	Albumin < 3.0 gm/dL
	Pre-albumin < 10 mg/dL
	Retinol binding protein < 10 μg/dL

Clinical definition of PIICS arranged by references [1,2,6].

**Table 2 jcm-09-02662-t002:** Basic characteristics with and without disseminated intravascular coagulation.

	ISTH Overt DIC	Non-ISTH Overt DIC	
***N***	**826**	**4571**	***p*-Value**
Age	73.5 ± 16.5	70.2 ± 18.5	<0.0001 *
Male, *n* (%)	499 (60.4)	2619 (57.3)	0.092
ICU on admission, *n* (%)	312 (37.7)	803 (17.6)	<0.0001 *
SOFA	7 (4, 10)	4 (2, 7)	<0.0001 *
APACHE II	18 (13, 25)	14 (10, 20)	<0.0001 *
Adjunctive therapy on admission			
Mechanical ventilation, *n* (%)	275 (33.3)	525 (11.5)	<0.0001 *
Duration (days)	4 (2, 10)	5 (2, 9)	0.36
Renal replacement therapy, *n* (%)	90 (10.9)	242 (5.3)	<0.0001 *
Duration (days)	4 (2, 9.5)	5 (2, 9)	0.32
Extracorporeal membrane oxygenation	11 (1.3)	22 (0.5)	<0.0040 *
duration (days)	4 (1, 5)	6 (3.75, 14)	0.019 *
Sepsis, *n* (%)	231 (28.0)	982 (21.5)	<0.0001 *
Post-surgery, *n* (%)	27 (3.3)	183 (4.0)	0.31
Laboratory findings on admission			
C-reactive protein, mg/dL	1.98 (0.2, 9.7)	0.8 (0.1, 5.4)	<0.0001 *
Albumin, g/dL	3.0 ± 0.9	3.4 ± 0.9	<0.0001 *
Lymphocyte counts, /μL	1093 (586, 2340)	1230 (720, 1890)	0.59

* *p*-value less than 0.05 as significant. ISTH overt DIC—International Society on Thrombosis and Haemostasis—overt disseminated intravascular coagulation; ICU—intensive care unit; SOFA—sequential organ failure assessment; APACHE II—acute physiology and chronic health evaluation.

**Table 3 jcm-09-02662-t003:** Outcomes including persistent inflammation, immunosuppression and catabolism syndrome (PIICS) with/without disseminated intravascular coagulation.

	ISTH overt DIC	Non-ISTH overt DIC	
***N***	**826**	**4571**	***p*-Value**
Mortality, *n* (%)	208 (25.2)	312 (6.8)	<0.0001 *
Length of hospital stay, days	5 (2, 11)	3 (2, 8)	<0.0001 *
Barthel index at discharge	60 (10, 100)	85 (30, 100)	<0.0001 *
Laboratory findings on day 14			
C-reactive protein (mg/dL)	3.6 (1.4, 7.2)	2.1 (0.6, 5.5)	<0.0001 *
Albumin (g/dL)	2.3 ± 0.8	2.5 ± 0.8	0.0015*
Lymphocyte (/μL)	900 (520, 1308)	1030 (665, 1405)	0.075
Outcomes, *n* (%)			<0.0001 *
PIICS	121 (14.7)	367 (8.0)	
Prolonged hospital stay without PIICS	41 (5.0)	204 (4.5)	
Early recovery	490 (59.3)	3758 (82.2)	
Early death	174 (21.1)	242 (5.3)	

* *p*-Value less than 0.05 as significant. ISTH overt DIC—International Society on Thrombosis and Haemostasis—overt disseminated intravascular coagulation; PIICS—persistent inflammation—immunosuppression and catabolism syndrome.

**Table 4 jcm-09-02662-t004:** Coagulation findings on admission and final outcomes.

	PIICS	Non-PIICS (Prolonged Hospital Stay without PIICS/Early Recovery)	Early Death	
***n***	**488**	**4493**	**416**	***p*-Value**
ISTH overt DIC (%)	121 (24.8) _a,b_	531 (11.8) _c_	174 (41.8)	<0.0001 *
JAAM DIC (%)	154 (31.6) _a,b_	618 (13.8) _c_	218 (52.4)	<0.0001 *
Platelet (×10,000/μL)	18.5 (12.4, 25.0) _d,e_	19.4 (14.9, 24.4) _f_	16.4 (10.4, 22.3)	<0.0001 *
FDP (μg/dL)	11.4 (5.7, 32.5) _d,e_	4.9 (2.5, 13.7) _f_	20.7 (6.4, 87.3)	<0.0001 *
Dimer (μg/dL)	7.6 (3.9, 19.1) _d,e_	3.8 (1.3, 9.5) _f_	12.9 (4.7, 50.8)	<0.0001 *
PT (%)	82.0 ± 25.9 _g,h_	92.0 ± 24.1 _i_	77.3 ± 29.6	<0.0001 *
APTT (s)	32.9 ± 15.3 _g,h_	30.6 ± 11.5 _i_	41.2 ± 27.3	<0.0001 *
Fibrinogen (mg/dL)	602.6 ± 267.2 _g,h_	488.8 ± 217.2 _i_	449.2 ± 219.3	<0.0001 *
Antithrombin activity (%)	79.7 ± 24.4 _g,h_	90.1 ± 26.1 _i_	73.7 ± 25.2	<0.0001 *

* *p*-value less than 0.05 as significant. (a) PIICS group is significantly different with non-PIICS group by Bonferroni test (*p* < 0.05/3); (b) PIICS group is significantly different with early death group by Bonferroni test (*p* < 0.05/3); (c) non-PIICS group is significantly different with early death group by Bonferroni test (*p* < 0.05/3).; (d) PIICS group is significantly different with non-PIICS group by Steel–Dwass test (*p* < 0.05); (e) PIICS group is significantly different with early death group by Steel–Dwass test (*p* < 0.05); (f) non-PIICS group is significantly different with early death group by Steel–Dwass test (p < 0.05); (g) PIICS group is significantly different with non-PIICS group by Tukey–Kramer test (*p* < 0.05); (h) PIICS group is significantly different with early death group by Tukey–Kramer test (p < 0.05); (i) non-PIICS group is significantly different with early death group by Tukey–Kramer test (*p* < 0.05). *p*-values are those of results obtained from one-way analysis of variance. PIICS—persistent inflammation—immunosuppression and catabolism syndrome; ISTH overt DIC—International Society on Thrombosis and Haemostasis—overt disseminated intravascular coagulation; JAAM DIC—Japanese Association for Acute Medicine—disseminated intravascular coagulation; FDP—fibrin/fibrinogen degradation products; PT—prothrombin time; APTT—activated partial thromboplastin time.

**Table 5 jcm-09-02662-t005:** Univariate/multivariate logistic regression analysis of persistent inflammation, immunosuppression and catabolism syndrome (PIICS).

	Univariate Logistic Regression Analysis		Multivariate Logistic Regression Analysis	
PIICS	Odds Ratio (95% CI)	*p*-Value	Odds Ratio (95% CI)	*p*-Value
Age	1.02 (1.01–1.03)	<0.0001 *	1.02 (1.01–1.04)	0.0038 *
Male	1.43 (1.17–1.74)	0.0004 *	2.10 (1.37–3.22)	0.0006 *
SOFA	1.23 (1.19–1.28)	<0.0001 *		
APACHEII	1.03 (1.02–1.04)	<0.0001 *	1.04 (1.01–1.06)	0.0012 *
Sepsis	3.70 (3.05–4.49)	<0.0001 *	1.60 (1.00–2.55)	0.049 *
C-reactive protein (mg/dL)	1.07 (1.06–1.08)	<0.0001 *	1.02 (0.99–1.04)	0.18
Albumin (g/dl)	0.57 (0.52–0.62)	<0.0001 *	0.60 (0.42–0.85)	0.0037 *
Lymphocyte counts (×1000/μL) _#_	0.93 (0.83–1.03)	0.15	1.19 (1.04–1.36)	0.011 *
Hemoglobin (g/dL)	0.95 (0.93–0.98)	0.0015 *	1.09 (1.00–1.19)	0.040 *
Creatinine (mg/dL)	1.11 (1.07–1.15)	<0.0001 *	1.11 (1.01–1.21)	0.034 *
HbA1c (%)	1.07 (0.99–1.14)	0.071	1.14 (1.01–1.31)	0.028 *
ISTH overt DIC	2.46 (1.96–3.08)	<0.0001 *	1.58 (1.02–2.46)	0.042 *
JAAM DIC	2.89 (2.35–3.56)	<0.0001 *		
Antithrombin activity (%)	0.98 (0.98–0.99)	<0.0001 *	0.99 (0.99–1.01)	0.81

*; *p*-value less than 0.05 as significant. (#) unit is changed from usual one based on the clinical insights. SOFA and JAAM DIC were excluded from the multivariable model because these variables had collinearity with APACHE and ISTH overt DIC, respectively. PIICS—persistent inflammation—immunosuppression and catabolism syndrome; ISTH overt DIC— oInternational Society on Thrombosis and Haemostasis—overt disseminated intravascular coagulation; JAAM DIC—Japanese Association for Acute Medicine—disseminated intravascular coagulation; SOFA—sequential organ failure assessment; APACHE—acute physiology and chronic health evaluation.

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
