# Peer review of "Disseminated Intravascular Coagulopathy Is Associated with the Outcome of Persistent Inflammation, Immunosuppression and Catabolism Syndrome"

_jcm, 2020, doi:10.3390/jcm9082662_

Round 1

Reviewer 1 Report

The present MS holds its importance for providing important information to clinicians and scientist working on the clinical studies on critical care patients and is well within the scope for this journal.

Strengths-

Methods are described with details

The study is designed well and conducted methodically

Limitations-

Language needs to be checked for clarity of meaning and grammatical mistakes.

Figure 2 needs to be described separately.

The MS is written in scientific manner though with complexity because of which the article may have limited readability and interest for common readers

Please highlight the first time findings coming out of this study.

The discussion part may be elaborated. 

Author Response

The present MS holds its importance for providing important information to clinicians and scientist working on the clinical studies on critical care patients and is well within the scope for this journal.

Strengths-

Methods are described with details

The study is designed well and conducted methodically

Our response: We would like to thank the reviewer for his/her review and appraisal of our manuscript. We revised it as the reviewer suggested.

Limitations-

Language needs to be checked for clarity of meaning and grammatical mistakes.

Our response: Revised manuscript was checked again by native English proof-reading service.

Figure 2 needs to be described separately.

Our response: We described the manuscript text and figure/figure legend separately.

The MS is written in scientific manner though with complexity because of which the article may have limited readability and interest for common readers

Our response: Thank you for this comment. We revised our manuscript to easy-to-understand one as possible.

Please highlight the first time findings coming out of this study.

Our response: We would like to thank this comment. This was the first study to clarify the correlation between coagulopathy and PIICS in the large clinical data. Their relation was described in discussion. We added the explanation in discussion as below.

“This report is the first of a study clarifying correlation between coagulopathy and PIICS based on large clinical data.”

The discussion part may be elaborated.

Our response: We additionally discussed about the correlations between each factors leading to PIICS development, obtained by our analysis, as below.

“In our generalized structural equation model, initial nutrition status, inflammation and coagulopathy affected PIICS development. It was reasonable because PIICS was fundamentally the inflammation and catabolism syndrome. These factors were mutually correlated through crosstalk between inflammation and coagulopathy. They were also affected strongly by the overall disease severity and infection condition. Sepsis has been reported as one of the strongest PIICS development conditions [1,2]. Furthermore, as our multivariable regression analysis results have suggested, the background of age, male sex and diabetes, and complication of renal dysfunction and anemia were associated with PIICS development. Comorbidities and higher age have been regarded as risk factors for PIICS [13]. Especially, kidney injury [3] and diabetes [23] were discussed as directly involved with PIICS development. Moreover, a recent study showed that anemia had been correlated with persistent inflammation in critically ill septic patients [24]. These comorbidities can be regarded as independent risk factors for PIICS.”

Reviewer 2 Report

Nakamura et al. present a retrospective investigation that implicates DIC as a contributor to persistent inflammation, immunosuppression and catabolism syndrome (PIICS).  The definition of PIICS presented by the authors as the presence of C reactive protein (CRP) >3.0 mg/dl or albumin <3.0 g/dl or lymphocyte count <800/μl on day 14 of hospitalization of critically ill patients.

The authors examined several thousand patients, and using a multifactorial model, determined that the occurrence of Japanese Association for Acute Medicine-disseminated intravascular coagulation (JAAM-DIC) was an independent risk factor for PIICS in 413 patients.

The manuscript has a number of serious issues.

First, as seen on page 2, the author discuss how they generated CRP values:

“Subsequently, CRP linear interpolation was performed for these other patients. If the patients met the following two conditions, then the evaluated CRP was more than 14 days: 1. The CRP was measured at more than two points during the hospital stay period. 2. The maximum CRP point in the first 7 days was designated as the starting point. CRP deficit points were linearly interpolated within 7 days. Linear interpolation was not applied if they did not meet these conditions.”

As for the other two possible criterion that were needed by day 14 (circulating albumin and lymphocytes), the authors permitted values obtained between day 11 and 17.

So, the major determinants of PIICS were either interpolated or approximated.  I do not accept interpolated data.  The authors provide no justification or citation to lead the readership to know that CRP concentrations either increase or decrease in a linear fashion during acute illness.  Given the nature of biological systems, I would not expect a linear relationship – people can become rapidly more or less inflamed.

I strongly suggest that the authors use the same 11-17 day approximation for CRP or remove patients that do not have this measurement within the time frame from analysis.  Then, with the likely decreased number of patients, perform the same statistical analyses.

With regard to statistics, the authors provide general ANOVA results without between group post hoc comparisons.  Such tests include Tukey or SNK.  Table 4 in particular has this problem.

Is blood purification hemodialysis?

The tables are not in order.

Table 5 basically shows that nearly all variates, even with an odds rations slightly above or below 1.0, are predictive of PIICS.  Even given the large number of patients per group, these significant odds ratios are clinically meaningless.  The authors have a clinical epidemiologist among them, but even this individual will have trouble explaining how an odds ratio of 1.00 (1.00–1.01) demonstrates a significant difference.  The authors are advised to obtain statistical assistance with their many and very complicated analyses.

If the authors can provide real, measured data and repeat this analysis, it will be of interest to the readership.

Author Response

Nakamura et al. present a retrospective investigation that implicates DIC as a contributor to persistent inflammation, immunosuppression and catabolism syndrome (PIICS).  The definition of PIICS presented by the authors as the presence of C reactive protein (CRP) >3.0 mg/dl or albumin <3.0 g/dl or lymphocyte count <800/μl on day 14 of hospitalization of critically ill patients.

The authors examined several thousand patients, and using a multifactorial model, determined that the occurrence of Japanese Association for Acute Medicine-disseminated intravascular coagulation (JAAM-DIC) was an independent risk factor for PIICS in 413 patients.

Our response: We would like to thank the reviewer for his/her careful review and important comments on our manuscript. We re-analyzed the dataset and revised the manuscript as the reviewer suggested.

The manuscript has a number of serious issues.

First, as seen on page 2, the author discuss how they generated CRP values:

“Subsequently, CRP linear interpolation was performed for these other patients. If the patients met the following two conditions, then the evaluated CRP was more than 14 days: 1. The CRP was measured at more than two points during the hospital stay period. 2. The maximum CRP point in the first 7 days was designated as the starting point. CRP deficit points were linearly interpolated within 7 days. Linear interpolation was not applied if they did not meet these conditions.”

As for the other two possible criterion that were needed by day 14 (circulating albumin and lymphocytes), the authors permitted values obtained between day 11 and 17.

So, the major determinants of PIICS were either interpolated or approximated.  I do not accept interpolated data.  The authors provide no justification or citation to lead the readership to know that CRP concentrations either increase or decrease in a linear fashion during acute illness.  Given the nature of biological systems, I would not expect a linear relationship – people can become rapidly more or less inflamed.

I strongly suggest that the authors use the same 11-17 day approximation for CRP or remove patients that do not have this measurement within the time frame from analysis.  Then, with the likely decreased number of patients, perform the same statistical analyses.

Our response: We would like to thank these important comments. We introduced CRP interpolation because CRP change would be delayed a little from a true inflammatory status and interpolation would complement it. However, we also agree with the reviewer and his/her comment ”people can become rapidly more or less inflamed”. As the reviewer suggested, we re-analyzed the dataset without CRP interpolation and permitted CRP values between day 11 and 17. There were 618 patients with CRP values in the 733 patients who stayed for more than 14 days with assignment as PIICS or prolonged hospital stay without PIICS. The patients who did not have CRP between day 11 and 17 could be roughly considered as no CRP-related PIICS because of it’s unnecessity. Along with albumin and lymphocyte counts, it was applied as negative in an or-conditional expression if they had no value. Re-analysis with them revealed that the results were identical with the previous one. We revised all the manuscript according to the statistical analysis with no CRP interpolation dataset.

With regard to statistics, the authors provide general ANOVA results without between group post hoc comparisons.  Such tests include Tukey or SNK.  Table 4 in particular has this problem.

Our response: Thank you for this important comment. We introduced Bonferroni, Steel-Dwass and Tukey-Kramer test for comparison between the groups in multigroup in table.4.

Is blood purification hemodialysis?

Our response: We apologize, that this blood purification included continuous and intermittent renal replacement therapy and did not include blood adsorption or plasma exchange. Therefore, we revised it as “renal replacement therapy”.

The tables are not in order.

Our response: We apologize, and revised it in order.

Table 5 basically shows that nearly all variates, even with an odds rations slightly above or below 1.0, are predictive of PIICS.  Even given the large number of patients per group, these significant odds ratios are clinically meaningless.  The authors have a clinical epidemiologist among them, but even this individual will have trouble explaining how an odds ratio of 1.00 (1.00–1.01) demonstrates a significant difference.  The authors are advised to obtain statistical assistance with their many and very complicated analyses.

Our response: We appreciate the reviewer's insightful comments. We consulted the statistician as the reviewer suggested. First, as pointed out, we have reanalyzed data using clinically-meaningful unit levels. For example, the unit of lymphocyte counts has been changed from /ul to X1000/ul based on the clinical insights. Second, in this reanalysis, we first performed univariate analysis for each risk factor. Then, we conducted a multivariable logistic regression model including all of these potential risk factors. The results of the reanalysis are shown in Table.5.

If the authors can provide real, measured data and repeat this analysis, it will be of interest to the readership.

Our response: We would like to thank this comment. We agree with the reviewer that we should conduct a prospective study in which day 14 data is fully analyzed in all the participants in further study. We also agree that it would be a limitation of our study. We added it in limitation section as below.

“Finally, CRP, albumin, and lymphocyte counts on day 14 were allowed within days 11–17, which included deficits. Laboratory data should be collected accordingly on day 14 without deficits for future prospective studies.”

Reviewer 3 Report

Authors retrospectively examined the relationship between PIICS and DIC.

Although this manuscript is potentially interesting, several issues arise.

JAAM-DIC is too high sensitive and low specific for DIC. I recommended to use ISTH-overt DIC or JMHLW diagnostic criteria.

Authors should show the merit of diagnosing PIICS.

Table 4. Authors should explain the statistical analyses.

Table 1. Does PIICS fill all items?

Abstract. The last sentence should be rewritten.

Authors should show the outcome of PIICS.  

Author Response

Authors retrospectively examined the relationship between PIICS and DIC.

Our response: We would like to thank the reviewer for his/her review and meaningful comments on our manuscript.

Although this manuscript is potentially interesting, several issues arise.

JAAM-DIC is too high sensitive and low specific for DIC. I recommended to use ISTH-overt DIC or JMHLW diagnostic criteria.

Our response: We would like to thank the reviewer for his/her important comment. We fully agree with this comment and used ISTH-overt DIC criteria in the revised manuscript. The results were identical with the previous one.

Authors should show the merit of diagnosing PIICS.

Our response: We described the merit of diagnosing PIICS in introduction as below.

“With such clinical criteria to detect PIICS development, the patients’ immunosuppression can be evaluated before actual infection occurs. The catabolism can be evaluated for eventual physical function and muscle volume.”

Table 4. Authors should explain the statistical analyses.

Our response: Thank you for this comment. We described the statistical analysis as below.

“a, PIICS group is significantly different with non-PIICS group by Bonferroni (p<0.05/3), Steel–Dwass (p<0.05) or Tukey–Kramer test (p<0.05).

b, PIICS group is significantly different with early death group by Bonferroni (p<0.05/3), Steel–Dwass (p<0.05) or Tukey–Kramer test (p<0.05).

c, non-PIICS group is significantly different with early death group by Bonferroni (p<0.05/3), Steel–Dwass (p<0.05) or Tukey–Kramer test (p<0.05).

p values are those of results obtained from one-way analysis of variance.”

Table 1. Does PIICS fill all items?

Our response: It was not described whether these conditions were needed to be filled “and” or “or” in the original references. However, prolonged stay and CRP would be most important for PIICS constitution as a persistent inflammation, and other factors could be “or” conditions, especially for many catabolism criteria. Therefore, we identified PIICS  as CRP >3.0 mg/dl or albumin <3.0 g/dl or lymphocyte counts of <800/μl in the patients whose hospital stay was more than 14 days. We described it in our revised introduction as below.

“The original reference materials do not describe whether these conditions needed to be filled in as “and” or “or”. However, prolonged stay and CRP are expected to be most important for PIICS leading to persistent inflammation [1,2]; other factors can be “or” conditions.”

Abstract. The last sentence should be rewritten.

Our response: We apologize. We revised the last sentence as below.

“Multivariate regression analysis and a generalized structural equation model identified DIC on admission as an independent risk factor for PIICS in surviving patients.”

Authors should show the outcome of PIICS. 

Our response: We described the miserable outcomes of PIICS in introduction as below.

“Once PIICS develops, other pathogens can easily infect the patient, activities of daily living (ADL) is worsen and their long-term mortality is eventually increased [1,2].”

Round 2

Reviewer 2 Report

Please only use one post hoc test, not three (tables 4 & 5).

Author Response

Please only use one post hoc test, not three (tables 4 & 5).

Our response: We used Bonferroni test for categorical variables, Steel–Dwass test for continuous and non-parametric variables, and  Tukey–Kramer test for continuous and parametric variables. We changed the explanation in table.4 as below. We did not use three tests in table.5.

a, PIICS group is significantly different with non-PIICS group by Bonferroni test (p<0.05/3).

b, PIICS group is significantly different with Early death group by Bonferroni test (p<0.05/3).

c, non-PIICS group is significantly different with Early death group by Bonferroni test (p<0.05/3).

d, PIICS group is significantly different with non-PIICS group by Steel–Dwass test (p<0.05).

e, PIICS group is significantly different with Early death group by Steel–Dwass test (p<0.05).

f, non-PIICS group is significantly different with Early death group by Steel–Dwass test (p<0.05).

g, PIICS group is significantly different with non-PIICS group by Tukey–Kramer test (p<0.05).

h, PIICS group is significantly different with Early death group by Tukey–Kramer test (p<0.05).

i non-PIICS group is significantly different with Early death group by Tukey–Kramer test (p<0.05).